# Health Impacts from Ambient Particle Exposure in Southern Sweden

**DOI:** 10.3390/ijerph17145064

**Published:** 2020-07-14

**Authors:** Ralf Rittner, Erin Flanagan, Anna Oudin, Ebba Malmqvist

**Affiliations:** 1Division of Occupational and Environmental Health, Lund University, 221 00 Lund, Sweden; erin.flanagan@med.lu.se (E.F.); anna.oudin@med.lu.se (A.O.); ebba.malmqvist@med.lu.se (E.M.); 2Occupational and Environmental Medicine, Department of Public Health and Clinical Medicine, Umeå University, 901 87 Umeå, Sweden

**Keywords:** health impact assessment, air pollution, mortality, burden of disease, low birth weight, HIA

## Abstract

A health impact assessment (HIA) is an important tool for making informed decisions regarding the design and evaluation of environmental interventions. In this study, we performed a quantitative HIA for the population of Scania (1,247,993), the southernmost county in Sweden, in 2016. The impact of annual mean concentrations of particulate matter with an aerodynamic diameter <2.5 µm (PM_2.5_), modeled at their home residences for the year 2011, on mortality, asthma, dementia, autism spectrum disorders, preeclampsia and low birth weight (LBW) was explored. Concentration–response (C-R) functions were taken from epidemiological studies reporting meta-analyses when available, and otherwise from single epidemiological studies. The average level of PM_2.5_ experienced by the study population was 11.88 µg/m^3^. The PM_2.5_ exposure was estimated to cause 9–11% of cases of LBW and 6% of deaths from natural causes. Locally produced PM_2.5_ alone contributed to 2–9% of the cases of diseases and disorders investigated. Reducing concentrations to a maximum of 10 µg/m^3^ would, according to our estimations, reduce mortality by 3% and reduce cases of LBW by 2%. Further analyses of separate emission sources’ distinct effects were also presented. Reduction of air pollution levels in the study area would, as expected, have a substantial effect on both mortality and adverse health outcomes. Reductions should be aimed for by local authorities and on national and even international levels.

## 1. Introduction

Air pollution is one of the leading environmental causes of morbidity and mortality worldwide, with more than 4 million premature deaths per year being attributed to outdoor air pollution [1]. Even in Nordic countries, despite relatively low levels of air pollution, air pollutants have been estimated to cause around 10,000 excess deaths every year [2]. Epidemiological evidence of associations between air pollution exposure and various adverse health effects has also been demonstrated in low exposure areas by the authors and others [3,4,5,6,7,8,9,10,11,12]. These adverse health effects can be both acute and chronic.

An air pollution health impact assessment (HIA) estimates mortality rates and incidence of disease based on current levels of air pollution as well as the subsequent public health impact following a change in air pollution concentrations. To facilitate these estimations, concentration–response (C–R) functions from the results of epidemiological studies are utilized. The quantitative part of the HIA results in the human health impacts of a proposed policy or an implemented action and facilitates dialogue with relevant stakeholders, such as policymakers and planners [13].

Vital to the HIA, the concentration–response function has traditionally been assumed to be the same for long-range transported and locally produced particulate matter with an aerodynamic diameter <2.5 µm (PM_2.5_) [14], most likely due to a lack of studies distinguishing between them. In countries such as Sweden, where in-transported PM_2.5_ is substantially larger than locally emitted PM_2.5_, this assumption may heavily affect the estimated health burden due to air pollution. However, recent research indicates that the concentration–response curve for PM_2.5_ from local sources might be considerably steeper than for regional background sources [15]. A potential explanation for this contrast might be that locally produced particles, being newly emitted, are more reactive.

The purpose of the present study was to estimate the quantitative health impacts of long-term exposure to PM_2.5_ in Scania, Sweden with special focus on premature mortality and incidence of select diseases for sensitive groups, including pregnant women and fetuses, children and the elderly. Moreover, the aim was to estimate these quantitative health impacts for both the total air pollution content and for specific sources, such as local traffic and small-scale heating, in order to provide more detailed input for policy actions. This expands upon our previous research [16], which was limited to the city of Malmö and investigated air pollution related to vehicle exhaust only.

## 2. Materials and Methods 

### 2.1. Study Setting and Study Population

Scania, the southernmost county in Sweden, spans 100 km^2^ and has a population of approximately 1.36 million. Here, annual mean concentrations of particulate matter with a diameter less than 2.5 µm (PM_2.5_) are relatively low compared to other regions of the world. Indeed, these levels readily comply with the European Union (EU) directive of 25 µg/m^3^ [17]. However, PM concentrations in Scania can at times exceed the more health-based threshold of 10 µg/m^3^ recommended by the World Health Organization (WHO), especially in urban areas.

The study population consisted of those aged 5 years and above residing in Scania at the end of 2016 with available geocodes for their residence (1,247,993 persons). These were the latest available data at the time of analysis. Region Skåne (a self-governing administrative region) records residential geographical coordinates at the end of each year. The study population’s demographics, including age and sex, diverge based upon their relevance to each health outcome investigated, as detailed below.

### 2.2. Health Outcomes

Mortality and incidence of disease, including asthma, preeclampsia, dementia and autism spectrum disorders as well as low birth weight (LBW), were explored in this study. While diverse, these health outcomes have been prevalent in epidemiological literature, with positive associations demonstrated for air pollution’s relationship to various respiratory diseases [18,19], pregnancy and birth outcomes [20], as well as neurological and cognitive disorders [21,22,23,24]. Furthermore, the Global Burden of Disease has evaluated enough scientific evidence to include LBW in their recent PM_2.5_ assessments [1]. For these reasons, we have chosen to include this broad range of health outcomes. A summary of these chosen health outcomes including population demographics and sources for baseline incidence can be found in Table 1.

For autism spectrum disorders (ICD-code F84), data were received from an ongoing cooperation with the Division of Children’s and Youth Psychiatry at Lund University. Approximately 1% of 48,571 children born in the catchment area of Lund and Malmö between 1999 and 2009 were diagnosed with autism spectrum disorders, where childhood autism (ICD-code F84.0) was the largest patient proportion (77%). 

Concerning LBW birth (defined as a birth weight of <2500 g) and preeclampsia, women in the age range of 15 to 44 (n = 244,458 with data available for analysis) were considered to be in their fertile life period. The rate of births in Scania is reported to be 64.5 per 1000 women (16,122 deliveries and 249,885 women), based on data taken from the Swedish Board of Health and Welfare [25]. This was calculated into the probability of each woman to give birth during 2016. The rate of LBW was reported to be 4.8%, and the prevalence of preeclampsia was 3.4% for the same year in Scania according to the medical birth register [26]. By combining the probability of giving birth with the probability of developing these conditions, we could determine the probability for each woman to have a baby with LBW or a preeclamptic pregnancy, which then was used as baseline data.

Baseline data on the remaining outcomes, asthma and dementia, were obtained from different sources. For children’s asthma, children aged 5 to 14 (n = 133,403) were included and an incidence of 0.01 per person from a Swedish study was used [27]. When calculating the HIA for dementia, all those of age 65 and above (n = 233,639) in Scania were included, and age-specific incidence for dementia was derived from an international study [28].

Finally, all Scanians aged 30 and older (n = 791,974) were included in the HIA for mortality. Baseline data on mortality for different age groups was provided by the Swedish Board of Health and Welfare, as further described in Malmqvist et al. [16].

### 2.3. Air Pollution Exposure

The exposure measure used was modeled annual concentrations of PM_2.5_ based on an emissions database, which consists of a large number of emission sources described by geographical location and relevant emission factors at the local level throughout the study area (Scania county) [29]. These data were the basis for a Gaussian dispersion calculation together with meteorological data, as further described by Rittner et al. [30]. In order to quantify the total concentration of PM_2.5_, it was necessary to account for background PM_2.5_ levels, which are not described in the emission database. These background levels were derived from measurements at a regional background monitoring station (Vavihill) and represent air pollution not significantly affected by local sources, but rather originating from long-range transport or produced by natural processes. For the population of Malmö, however, an urban background monitor located in the city (roof of Malmö City Hall) was considered better than the rural Vavihill monitor. This is because Malmö is situated close to Denmark and Öresund, where significant levels of air pollution are produced and brought into southwestern Scania by westerly winds. In this case, the background levels used were measurements from the Malmö City Hall with the modeled local concentrations subtracted.

We studied the effect of total concentration levels of PM_2.5_ (locally emitted plus background) as well as only local pollution levels. The composition of the modeling software also enabled the selection of specific emission sources at the local level, such as traffic and small-scale heating, to be investigated separately. We used exposure calculations from the year 2011 as it represents a “normal” year regarding weather patterns. In selecting a stable year, we minimize the possibility of unusual or extreme weather events influencing exposure data, such as the quantity of in-transported air pollutants.

Each person was assigned the PM_2.5_ concentration value of a 100 × 100 m cell in which the centroid of their residence at the end of 2016 was located.

Table 2 illustrates the specific exposure scenarios used for the various health outcomes under investigation.

### 2.4. Health Impact Analyses

We used C–R relationships for the described health outcomes according to methods applied in Gustafsson [31] and Malmqvist et al. [16]. The former involved exposure assessments on a national level in Sweden, while the latter estimated the health benefits of exhaust-free transportation in the city of Malmö. These studies assessed exposure differently; however, both include health impact calculations based on knowledge of baseline risk (incidence rate) for the outcome, the extent of exposure and an accurate C–R function (sometimes also referred to as exposure–response function).

As LBW was not included in the studies named above, we used a review article by Perera et al. for the C–R relationship [20]. In addition, autism spectrum disorders were analyzed with the C–R based on Becerra et al. [21]. C–R functions for mortality were derived from Turner et al. [15], where different hazard ratios (HR) for regional concentrations and near-source emissions are recommended. In addition to the C–R function itself, the endpoints of each C–R’s 95% confidence interval (CI) was applied and calculated. All C–R functions and their respective 95% CIs, their sources and their reported exposure ranges are listed in Table 1.

The health impact function (HIF) below can be derived from the population attributed fraction (PAF) definition, as described by Martenies et al. [32]. The PAF, as Martenies et al. say, considers the probability for the whole population to be exposed, while the HIF estimates the change in individual exposure and outcome incidence. By employing an HIF, we utilized the exposure for each study person and their baseline incidence according to actual age and sex. The formula presented below was calculated for each person, and the individual changes in outcome were summed to obtain the total change for the relevant population selection. We have used a log-linear C–R estimate, resulting in the following form of the HIF [32]:ΔY = Y_0_ × (1 − e^−β × Δx^)(1)
where ΔY is the change in health outcome, Y_0_ is the baseline incidence of cases, (β) is the exposure (C–R) association (log relative risk) multiplied by the increase in exposure and (Δx) is the change in air pollution level (on an individual level). By inserting the different PM_2.5_ concentrations as (Δx), the health impact attributable to an observed situation or a reduction by intervention can be calculated.

For this study, we used the following hypothetical targets: removing (1) all PM_2.5_ air pollution, (2) all local PM_2.5_ air pollution, (3) PM_2.5_ air pollution from local traffic and (4) PM_2.5_ air pollution from local small-scale heating, as well as complying with (5) WHO emission guidelines and (6) United States Environmental Protection Agency (EPA) targets. See Table 2 for these hypothetical scenarios and the health outcomes considered for each.

As a sensitivity analysis, we also compared the more traditional method of using the same C–R for both local and long-range emissions to the more recently proposed approach of using different C–R functions for each [15].

### 2.5. Software Used

The dispersion modeling (Aermod-based implementation) was run in the commercial ENVIMAN module AQplanner. ENVMAN also provides a module for managing the emission database and necessary meteorological data. A self-written C program extracted the monthly means of PM_2.5_ at the grid-square centroids from binary dispersion result files. ArcGis 10.3 was used to import PM_2.5_ concentrations, represent them in a geographical grid, and extract the specific concentration value at each study person’s residence point. IBM SPSS Statistics 26 (IBM^®^, Armonk, NY, USA) was then used to calculate the final HIF.

## 3. Results

In this estimation, the mean of all individuals’ residential annual average concentrations of total PM_2.5_ for the Scanian population in 2011 was 11.88 µg/m^3^ and ranged from 9.56 to 19.4 µg/m^3^. Only a small portion of this value, 0.88 µg/m^3^, comprises locally emitted PM_2.5_. Regarding source-specific contributions to local PM_2.5_, also expressed as annual means, traffic accounted for 0.26 µg/m^3^, small-scale heating 0.19 µg/m^3^ and shipping and industry 0.02 µg/m^3^ each (Appendix A). See also Appendix A for exposure levels further divided by population subgroups and type of city, including a large city, a harbor city, and a municipality where small-scale heating is used frequently (heating municipality).

### 3.1. Mortality

Traffic is attributable to a larger proportion of deaths than small-scale heating (0.6% vs. 0.4%), given the assumption that the two sources of PM_2.5_ have equal effects (Table 3). As shown in Table 3 and Table 4, 219 deaths (2% of all deaths) could be avoided if all local PM_2.5_ was removed, again, assuming equal effects for all local sources. Moreover, reaching WHO’s air quality guideline of 10 µg/m^3^ or the United States EPA’s standard of 12 µg/m^3^ as a maximum annual average for each person at their residence would result in the reduction of mortality by 75 or 47 further deaths, respectively, in addition to the previously mentioned 219 when removing only concentrations from local emission sources; see Table 4.

### 3.2. Incidence of Diseases and Disorders

Our calculations demonstrate that PM_2.5_ air pollution in Scania affects the health of various vulnerable groups (see Table 3). Locally emitted PM_2.5_ in particular accounts for about 2% of childhood asthma cases and 3% of autism spectrum disorder (mainly autism) diagnoses. Among pregnant women, local PM_2.5_ was found to cause 5% of preeclampsia cases. Within the elderly population, 1% of dementia cases could be attributed to these local PM_2.5_ concentrations. Considering total PM_2.5_, this exposure accounted for 11% of LBW instances compared to baseline. Moreover, if a maximum annual mean of 10 µg/m^3^ for each woman was reached, 15 LBW incidences could be avoided (see Table 4).

### 3.3. Sensitivity Analysis

The contribution of local emissions is attributable to 2% of all mortality cases when applying the more recent C–R function. However, this contribution was only 0.5% (not shown in table) when calculated in a more traditional way: using the same C–R for local emissions as for long-range background emissions. Considering the total concentration of PM_2.5_ (regional background plus local contribution), the results from using the two methods became more equal. That is, total PM_2.5_ accounted for 7% of mortality cases when using the same function for both pollution categories and 6% (as shown in Table 3) when using a steeper curve (stronger effect) for local emissions.

## 4. Discussion

Despite Scania having population annual mean residential levels of PM_2.5_ around 12 µg/m^3^, which is well below the current EU regulation of 25 µg/m^3^ and their proposed regulation of 20 µg/m^3^, we estimated substantial health impacts. The effect of PM_2.5_ varied by health outcome with around 11% of all LBW cases and 6% of premature deaths being attributed to PM_2.5_. Using dose–response curves from Turner et al. [15] that distinguish between local and regional pollutants, we estimated that local PM_2.5_ emissions contribute to 2% of deaths related to baseline, while regional background emissions contribute to 4%. Regarding other health effects, local emissions were seen to contribute to 2% up to 9% of the incidence in cases at baseline. A reduction of local emissions, the most feasible course of action for local authorities, would thus have a substantial impact on the studied health outcomes as well as all-cause mortality (see Table 3). 

We present our study as a quantitative HIA from different sources, both local, total and divided by local emission sources, but the results can also represent the burden of disease from these sources. Our results indicate that the removal of all local PM_2.5_ sources would not suffice in reaching potential maximum concentrations of 10 or 12 µg/m^3^ adopted by the WHO and the United States EPA, respectively. As illustrated in Table 4, the decline of both mortality and LBW outcomes seems to be greater when these air quality targets are met than when only locally emitted PM_2.5_ concentrations are removed. With this, the reduction of background PM_2.5_ levels is also necessary to achieve these international air quality guidelines and for all inhabitants in Scania to subsequently enjoy their related health benefits.

Our investigation of different PM_2.5_ sources was not as thorough as Segersson et al. [3], where authors report traffic exhaust, traffic wear, residential wood combustion, shipping and other. However, we can still compare our findings on premature deaths attributed to local versus long-range transported PM_2.5_. For instance, Segersson et al. report the percentage of premature deaths attributed to local air pollution sources to be 50–70% in Stockholm, 46–69% in Umeå and 58–77% in Gothenburg. The reported ranges comprise the relative risks from different studies. Our findings demonstrate local traffic, small-scale heating and all local PM_2.5_ sources causing 9% (63 deaths), 7% (49 deaths) and 31% (219 deaths) of premature deaths in Scania, respectively (see Table 3). The differing impacts of local PM_2.5_ seen from city to city and for Scania as a whole can be explained by a number of factors. For example, Stockholm is more densely populated than Scania, where the population is spread over several cities. Additionally, we used more recently published C–R functions based on Turner et al. [15], whereas Segersson et al. used older reference materials including Jerrett et al. [36], Hoek et al. [37] and Janssen et al. [38]. The location of Scania close to the European continent where westerly winds carry air pollution produced elsewhere into the region likely contributes to our findings of local sources contributing to only 7.4% of the population’s mean exposure compared to the long-range transported PM_2.5_ (0.88 vs. Eleven µg/m^3^, respectively), see Appendix A. Understandably, this long-range transported PM_2.5_ contributed to a higher proportion of mortality cases in Scania, comprising 79%% (480 deaths). As a comparison to our Scanian levels, Segersson et al. found a total population weighted PM_2.5_ of 6.5 µg/m^3^, while they found a sum of local PM_2.5_ of 1.9 µg/m^3^ (29% of 6.5 µg/m^3^) in Stockholm.

Since only locally emitted PM_2.5_ concentrations, constituting 7.4% of total concentrations expressed as population residential mean of annual averages, were considered for the calculation of adverse health outcomes, it would be reasonable to assume that the impact on incidence of the investigated diseases and disorders would have been substantially higher if background levels had also been included in the analysis. However, long-range transported pollutants would not be affected by local interventions or policies, which is why we focused solely on locally emitted particles. 

### 4.1. Background PM_2.5_

The speciation and origin of particles at Vavihill regional background site has been studied by Genberg et al. [39] and was found to change significantly with the seasons. During the summer, particle levels are generally lower and are dominantly of biogenic origin, with 80% of the total carbon being organic carbon. Throughout the winter, however, biomass combustion and fossil fuel combustion were the main contributors, accounting for 32% and 28% of total carbon in the carbonaceous aerosol, respectively.

Particle levels and content at Vavihill vary in shorter time frames too, which is largely affected by where the air mass has originated or traveled from [40]. For example, our study area neighbors Germany and Poland, where, according to data from the European Monitoring and Evaluation Programme (EMEP), a large proportion of European coal power is produced. Coal power plants contribute substantially to the formation of PM_2.5_ through their emissions of sulphur dioxide (SO_2_) and nitrous oxides (NO_x_), which react with ammonia to form PM_2.5_ in the atmosphere. Given the large distance particles can travel [40], the Scanian population’s exposure and, consequently, their health are affected by the energy policies and actions of nearby countries. 

### 4.2. Reduction of Local PM_2.5_

As it is difficult and complex to reduce background PM_2.5_ concentrations, it is important to continue to reduce local emissions. The number of studies separating health effects of local, near-source, and background, sometimes referred to as regional, emissions are still few. However, differences in health effects attributed to local and regional PM_2.5_ was illustrated by Turner et al. [15]; authors reported a hazard ratio of 1.26 (95% CI: 1.19–1.34) in relation to each 10-unit increase in near-source PM_2.5_ compared to 1.04 (95% CI: 1.02–1.06) for regional PM_2.5_. Daily variations of local PM_2.5_ have been demonstrated to have a substantial impact on the number of daily deaths [12], where a change in the PM_2.5_ interquartile range was seen to be associated with a 0.90% increase in mortality (95% CI: 0.25–1.56%). Moreover, in a previous study [16], we estimated the hypothetical health gain from eliminating traffic exhaust emission in the city of Malmö. Here, we found that the number of deaths attributed to local traffic emissions was much higher than the number of deaths from traffic accidents (64 versus 8 deaths).

Our source-specific analysis showed rather small effects for both traffic and small-scale heating on mortality (see Table 3). As seen in Appendix A, the prominence of a PM_2.5_ source can vary depending on the type of city and its contextual factors. For instance, small-scale heating typically constitutes a small portion of the local PM_2.5_, except in municipalities that tend to have smaller and less dense populations, such as Osby, where small-scale heating is more prevalent. Similar trends are seen in the harbor cities, like Trelleborg, where the contribution of local PM_2.5_ emissions from shipping are higher than in other cities. Further, traffic becomes the largest source of mean exposure to local PM_2.5_, reaching approximately one-third of the total, only in Scania’s major city, Malmö. With these observations, it becomes clear that interventions aiming to reduce health impacts from particle emissions must be based on the local context. Regulating traffic, for example, would have a greater impact in larger cities, while more efficient wood stoves and heaters would be more effective in heating municipalities. Knowledge about the composition and context of cities on a lower level, as opposed to blanketed interventions at the national level, is necessary for successful air quality improvement and its subsequent health-related benefits. 

### 4.3. Methodological Considerations 

We consider the high-spatial-resolution modeling of PM_2.5_ exposure to be a major strength of this study. Additionally, using health baseline data from high-quality national registers or international studies also added to the reliability of the presented results. Utilizing the total population of our study area (Scania, Sweden) and having individual data on each resident’s age, sex and home coordinate were also important strengths. As the results of this study are based on the Scanian population’s demographic composition, the generalizability of our findings to other, diverse populations may be limited.

Moreover, the home residence is not where most people spend the majority of their time, and modeled levels outside a person’s home do not necessarily correlate very well with actual exposure [41]. Despite being a potential cause of exposure misclassification, this corresponds to the standard methods of epidemiological studies upon which the C–R functions are based.

The possibility of exposure misclassification also arises from our PM_2.5_ concentrations being derived from database-driven dispersion modeling as opposed to individual monitoring measurements, the latter not being feasible for the region’s total population (n = 1,247,993). However, the modeling was evaluated against independent air pollution monitoring stations where actual measurements were recorded with satisfying correlations [30]. Importantly, the best measurement-to-modeling correlation was seen in the southwest part of the county, which is where a large proportion of Scania’s population resides [42]. We considered both instances of exposure misclassification to be non-differential and, therefore, not likely to bias the results in any direction.

Limitations also exist with regard to the C–R functions available. For instance, C–R functions for two health outcomes, dementia and autism spectrum disorder are more uncertain. Thus, results pertaining to these adverse health outcomes may be less reliable and should be interpreted with caution.

Concerning C–R functions and exposures, it could be argued that C–R functions vary for particles with different chemical constitutions. In a Swedish study, for example, Ljungman et al. [43] demonstrated that health effects differed between source contributions to PM and black carbon (BC) from road wear, traffic exhaust and residential heating. Another study indicates that the association between PM and lung cancer may be attributed to various PM sources and their components [44]. Yet because epidemiological evidence is still limited, C–R functions for small-scale heating and traffic were assumed to be identical in our study. The sensitivity of which C–R function to use was illustrated by the results of Segersson et al. [3] and also Malmqvist et al. [45]. Unfortunately, the only descriptive data available on PM in our study were that its aerodynamic diameter was less than 2.5 µm; therefore, we could not study the possible effects of particles’ variation in chemical constitution.

As mentioned, previously, a large uncertainty exists in the accuracy of using a single C–R function for both locally produced particles and regional background particles. We have chosen to use C–R functions from Turner et al. [15], which differentiate the two origins of PM emission, but it is uncertain if these apply to our setting. Turner et al. used a nationwide (United States) Land Use Regression-odel with near-source PM_2.5_ within 1 km. Conversely, our study was based on dispersion calculations for 100 × 100 m grid cells surrounded by 2 levels of coarser grids covering large areas of Scania [30].

Another methodological approach would be to use national C–R functions for all outcomes, as was done in Denmark when Bronnum-Hansen et al. [46] studied the health effects of nitrogen dioxide (NO_2_). Authors were able to derive their C–R functions from the same population, as investigated, by having long-time NO_2_ concentrations in the same area for the same population. This method was, unfortunately, not possible to implement due to the scope of our study. Bronnum-Hansen et al. also stated that their exposure–response estimates for mortality were similar to those based on the meta-analyses from cohort studies, our approach, as reported by Faustini et al. [47].

Our study was conducted in collaboration with stakeholders such as the Swedish EPA and Region Skåne and, thus, contributes to the participatory approach for conducting HIAs as recommended by Nieuwenhuijsen et al. [48].

### 4.4. Future Research 

To conduct a relevant HIA of high quality, choosing an appropriate C–R function is essential. Given that there are very few epidemiological studies where the health effects of local and regional particles are investigated separately, it would be of significant interest to conduct more such studies. This is especially important considering the possible differences in effect that long-range transported (aged, less volatile) and locally emitted (younger, possibly more reactive) particles may have. Additionally, we believe that further research is necessary to establish the correct C–R functions for different air pollution sources and compositions. While we covered a large scope of air pollution’s adverse health effects, including respiratory illness, pregnancy conditions, cognitive function and mortality, there are countless other health outcomes and metrics to examine, such as disability-adjusted life years (DALYs) and the economic consequences of sick leave taken due to air pollution-related illness. Furthermore, it is important to have a holistic perspective and investigate other, concurrent health impacts when allocating resources and planning interventions to improve public health. For example, the health impacts of a reduction in greenhouse gases, increased physical activity (as a result of more active commuting) and expected changes in traffic accidents could be taken into account to more fully understand the consequences of interventions, for example, initiatives aiming to reduce traffic-related air pollution [49].

As mentioned above, future studies should corroborate their findings in order to better study the health effects of locally emitted particles versus particles stemming from long-range transport.

## 5. Conclusions

In conclusion, our findings indicate that reduction of PM_2.5_ emissions would have a substantial impact on both mortality and adverse health effects throughout the lifecycle yet to varying degrees. While it may be difficult for local and regional policymakers to reduce background PM_2.5_ concentrations, the potential to further reduce local emissions remains. As such, reductions are beneficial for the health and well-being of the Scanian population, and policies on air quality improvement should be a priority for authorities at all levels.

## Figures and Tables

**Table 1 ijerph-17-05064-t001:** Epidemiological associations underlying concentration–response (C–R) functions used in the calculations for this HIA, descriptive and source information and additional health outcome incidence sources.

Outcome	Demographic Specifications	C–R Function Based on (95% CI)	PM_2.5_ Range or Mean (SD) (µg/m^3^)	Source for C–R	Source for Baseline Incidence
**Mortality** **(all-cause)**	≥30 years old	HR 1.26 (1.19–1.31)	1.4–27.9	[15] ^1,7^	Board of health and welfare, Sweden
		per 10 µg/m^3^	12.6 (2.9)		
		HR 1.04 (1.02–1.06)	1.4–27.9	[15] ^1,8^	
		per 10 µg/m^3^	12.6 (2.9)		
**Dementia**	≥65 years old	HR 1.04 (1.03–1.05)	1.1–49.7	[33] ^2^	International study [28]
		per 4.8 µg/m^3^	^9^		
**Asthma**	5–14 years old	OR 1.03 (1.01–1.05)	0.4–3.34	[18] ^3^	Swedish study [27]
		per 1 μg/m^3^	^9^		
**Preeclampsia**	Females 15–44 years old	OR 1.31 (1.14–1.50)	10.1–17.3	[34] ^4^	Local birth register [35]
		per 5 µg/m^3^	^9^		
**Autism spectrum disorders**	Children	OR 1.15 (1.06–1.24)	^9^	[21] ^5^	Local study ongoing
		per 4.68 μg/m^3^	19.6 (3.5)		
**Low birth weight**	Births of females 15–44 years old	OR 1.09 (1.03–1.15)	1.4–98.1	[20] ^6^	Board of health and welfare, Sweden [26]
		per 10 µg/m^3^	^9^		

^1^ Single study with long follow up.^2^ Single study. ^3^ Meta-analysis of 40 eligible studies. ^4^ Meta-analysis of 4 studies. ^5^ Single study. ^6^ Meta-analysis of 19 studies recommended in a review of 5 studies. ^7^ Local source emissions ^8^ Long-range transported emissions ^9^ Not reported

**Table 2 ijerph-17-05064-t002:** Air pollution scenarios analyzed: Removal of various PM_2.5_ sources and the removal of PM_2.5_ concentrations exceeding the United States EPA and WHO air quality targets.

Outcome	Local and Background PM_2.5_	All Local PM_2.5_	Traffic PM_2.5_	Small-Scale Heating PM_2.5_	Maximum 12 µg/m^3 1^ of PM_2.5_	Maximum 10 µg/m^3 2^ of PM_2.5_
Mortality	X	X	X	X	X	X
Diseases/disorders		X				
LBW	X	X			X	X

^1^ United States EPA standard as maximum for each person; ^2^ WHO guideline as maximum for each person.

**Table 3 ijerph-17-05064-t003:** Health impacts from various PM_2.5_ sources using modelled concentrations for 2011 applied to Scania’s 2016 population.

Outcome	Baseline N	Cases Attributable to Exposure (% of baseline N)	Confidence Interval ^7^
**Local PM_2.5_ all Sources**			
Mortality ^1^	10,987	219 (2)	164–256
Dementia ^2^	10,260	66 (1)	50–82
Asthma ^3^	1330	33 (2)	11–55
Preeclampsia ^4^	536	28 (5)	13–43
Autism spectrum disorders ^5^	2810	79 (3)	29–109
Low birth weight ^6^	757	6 (0.8)	2–10
**Local PM_2.5_ Traffic**			
Mortality ^1^	10,987	63 (0.6)	47–73
**Local PM_2.5_ Small-scale Heating**			
Mortality ^1^	10,987	49 (0.4)	37–58
**Total PM_2.5_ (Local plus Regional Background)**			
Mortality ^1,^*	10,987	699 (6)	405–903
Low birth weight ^6^	757	84 (11)	28–142

^1^ Concentration–response function(s) from Turner et al. [15], * both the local and the regional component of concentration–response function used; ^2^ estimate 1.04 per 4.8 µg/m^3^ increased local particulate matter with an aerodynamic diameter < 2.5 µm(PM_2.5_) (Chen et al. [33]); ^3^ 1.03 (1.01–1.05) per 1 μg/m^3^ increased PM_2.5_ (Khreis et al. [18]); ^4^ OR 1.31 (1.14–1.50) per 5 µg/m^3^ increased PM_2.5_ (Pedersen et al. [34]); ^5^ OR 1.15 per 4.68 μg/m^3^ increased PM_2.5_ (Becerra et al. [21]); ^6^ OR 1.09 (1.03–1.15) (Perera et al. [20]); ^7^ Based on CI for C–R reported in literature source.

**Table 4 ijerph-17-05064-t004:** Change in the number of outcomes (LBW instances or deaths) in hypothetical scenarios involving the reduction of PM_2.5_ concentrations to various degrees.

	Reduction to Maximum 12 ^1^ µg/m^3^	Reduction to Maximum 10 ^2^ µg/m^3^	Removal of Local PM_2.5_
LBW	−10	−15	−6
Mortality	−266	−294	−219

^1^ United States EPA standard as maximum for each person; ^2^ WHO guideline as maximum for each person.

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
