# Peer review of "Health Impacts from Ambient Particle Exposure in Southern Sweden"

_ijerph, 2020, doi:10.3390/ijerph17145064_

Round 1
Reviewer 1 Report
General comments
Thank you for your interesting study on the health burden of residential PM2.5 and health impact assessment of changes in exposure to PM2.5 for the Scania area in Sweden. This type of studies are very much needed to further support the evidence of the damaging effect of air pollution, even at low levels. The study is very well written, and clear. My comments below address the use of definitions and request for more details for data inputs.
- The authors frame the analysis as health impact assessment, however, one component, where they estimate total elimination, aligns more to a burden of disease study, and then the assessment of alternative scenarios seems the health impact assessment component. The authors may choose to ignore this comment.
- I suggest that the authors clarify that this is a quantitative health impact assessment as health impact assessment is a process that includes a quantification of health impacts, amongst many other steps.
- I suggest adding to table 1 whether the RRs used are from meta-analysis or single studies. Also, the introduction (line 17) uses the term reviews. Are these studies meta-analyses?
- As to RRs, can you please specify the range of exposure in the source studies?
- In line 125 there is reference to two studies for the application for the health impact analysis methods. Can you please briefly explain what these methods are?
- Can you please specify early on in the text how morbidy was measured? Later on it is specified as incidence. Also, related to this, were the RRs chosen specific to the model outcomes mortality and incidence?
- Can you please specify software or programming language use for both analyses of air pollution and health impacts.
- I recommend that you add uncertainty analysis, at least for the input parameters for the health impact (e.g. RRs).
- Inputs for exposure and health impact are mentioned along the text. A table summarising all inputs would be useful.
Reviewer 2 Report
This article raises very important issues regarding the impact of air pollution on people's health and lives. The presented research results show that despite the low concentration of pollutants, specifically analyzed in the work of PM2.5, they have an impact on the increase in morbidity and mortality. This is particularly important for the most vulnerable people, i.e. children, the elderly and pregnant women. By choosing such topics, the authors presented a very high level of awareness of the impact of even a small pollution on the basic right people, which is human life and health. Analysis of the impact of PM2.5 on the health and life of residents from contaminated areas is all the more important as such small dust particles, and in particular their chemical composition, can have a disastrous effect on our health. Therefore, as the authors mentioned, it would be important to analyze the chemical composition of PM2.5 during future studies.
Due to the fact that the calculation methodology is based primarily on earlier work, it would be worth describing it in more detail in the description of the methodology at work.
As for the editorial page of the work, you definitely need to correct the page numbering (which has been mixed up due to the insertion of one page horizontally). In addition, I am convinced that Table 1 may be presented vertically, after minor editorial changes.
Also, in verse 135, the entry "Y0" is different than in the formula above (1), "0" was not written as a subscript.
Author Response
Pleas see the attachment.

Reviewer 3 Report
This is a very interesting study and a well-written paper.
I think that the analysis and tables could be more clarified.
